# The Effect of Hydroxytyrosol in Type II Epithelial-Mesenchymal Transition in Human Skin Wound Healing

**DOI:** 10.3390/molecules28062652

**Published:** 2023-03-15

**Authors:** Wafa Ali Batarfi, Mohd Heikal Mohd Yunus, Adila A. Hamid

**Affiliations:** Department of Physiology, Faculty of Medicine, Universiti Kebangsaan Malaysia, Jalan Yaacob Latiff, Bandar Tun Razak, Kuala Lumpur 56000, Malaysia

**Keywords:** wound healing, epithelial-mesenchymal transition type II, human dermal fibroblasts, hydroxytyrosol

## Abstract

Skin wound healing is a multiphase physiological process that involves the activation of numerous types of cells and is characterized by four phases, namely haemostasis, inflammatory, proliferative, and remodeling. However, on some occasions this healing becomes pathological, resulting in fibrosis. Epithelial mesenchymal transition (EMT) is an important process in which epithelial cells acquire mesenchymal fibroblast-like characteristics. Hydroxytyrosol (HT) is a phenolic compound extracted from olive oil and has been proven to have several health benefits. The aim of this study was to determine the effect of HT in type II EMT in human skin wound healing via cell viability, proliferation, migration, and proteins expression. Human dermal fibroblasts (HDF) isolated from skin samples were cultured in different concentrations of HT and EMT model, induced by adding 5 ng/mL of transforming growth factor-beta (TGF-β) to the cells. HT concentrations were determined via 3-(4,5-dimethylthiazol-2-yl)-2,5-diphenyltetrazolium bromide (MTT) assay. Cells’ migrations were evaluated using scratch and transwell migration assay. Protein expressions were evaluated via immunocytochemistry. The result showed that HT at 0.2% and 0.4% significantly increased the proliferation rate of HDF (*p* < 0.05) compared to control. Scratch assay after 24 h showed increased cell migration in cells treated with 0.4% HT (*p* < 0.05) compared to the other groups. After 48 h, both concentrations of HT showed increased cell migration (*p* < 0.05) compared to the TGF-β group. Transwell migration revealed that HT enhanced the migration capacity of cells significantly (*p* < 0.05) as compared to TGF-β and the control group. In addition, HT supplemented cells upregulate the expression of epithelial marker E-cadherin while downregulating the expression of mesenchymal marker vimentin in comparison to TGF-β group and control group. This study showed that HT has the ability to inhibit EMT, which has potential in the inhibition of fibrosis and persistent inflammation related to skin wound healing.

## 1. Introduction

Skin wound healing is a multiphasic extraordinary physiological process involving the activation of numerous types of cells, growth factors, and cytokines, and is characterized by four phases: hemostasis, inflammatory, proliferative and remodeling [1,2]. In the haemostatic phase, the damaged cells release several signals, which activate platelets and start the clotting cascade [3]. Activated platelets release many factors that act as chemoatttractants for cells involved in the inflammatory phase [4]. In response to the platelet’s signals, the inflammatory phase is started by migrating neutrophils to the wound site to debride the injured tissue and eliminate bacteria by producing a wide range of highly active antibacterial substances and proteases. After a few days, neutrophils leave the wound site, followed by macrophage migration into the wound site to phagocyte dead neutrophils [5,6]. Dermal fibroblast cells play a critical role in the proliferative phase by restoring the dermal layer by repopulating the wound bed, dissolving the temporary fibrin matrix through the secretion of matrix metalloproteinase (MMP), and finally differentiate into myofibroblasts to promote wound closure [5,7,8]. At the same time, new blood vessels proliferate throughout all phases of wound healing. Remodeling is the final phase of wound healing that is concerned with keeping the ECM’s degradation and synthesis in balance. The ECM degradation is caused by matrix metalloproteinase (MMP), which is produced by neutrophils, macrophages, and fibroblasts in the wound [9]. MMP activation and inhibition must be balanced until complete healing is accomplished [10].

The molecular actions surrounding the wound healing process have been linked to epithelial-mesenchymal transition (EMT) [11]. EMT is an important process in which epithelial cells acquire mesenchymal fibroblast-like characteristics such as reduced intercellular adhesion and enhanced motility, with loss of their epithelial functions and characteristics [12]. Based on biological context, EMT is frequently divided into three distinctive types: type I, which occurs during embryogenesis and organ development; Type II, which happens in tissue repair and wound healing process; and Type III, which is related to the metastatic progression of malignancy [13]. However, all three types of EMT have the same result: mobile cells of mesenchymal phenotype are produced from adhering epithelial cells with apical-basal polarity [14]. The loss of epithelial cell markers is the initial step in EMT, with decreased E-cadherin expression being one of the most significant. E-cadherin is responsible for sustaining epithelial cell lateral connections via adherent junctions, as well as cell adhesion and relative immobility in the tissue [15,16].

Several growth and differentiation factors can initiate or regulate EMT, including transforming growth factor-beta (TGF-β), fibroblast growth factor (FGF), hepatic growth factor (HGF), platelet-derived growth factor (PDGF) and Wnt and Notch proteins [17]. TGF-β is a multifunctional cytokine that causes a variety of cellular processes such as growth inhibition, tissue fibrosis, and EMT. The activation of TGF-β signaling pathways occurs when TGF-β binds to its receptors, namely TGF-β type II (TβRII) and type I (TβRI) serine/threonine kinase receptors. TβRII binds to TGF-β, which transphosphorylates TβRI, and then activates R-Smads (Smad2 and Smad3) by phosphorylating their C-terminal serine residues. R-Smads that have been activated form a heterocomplex with Smad4 and translocate into the nucleus to control gene expression [18]. These pathways are named Smad-dependent pathways. However, TGF-β activates a variety of non-Smad signaling pathways as well. TGF-β induced EMT has been associated with activating Ras, mitogen-activated protein kinases (MAPKs) such as ERK and p38 MAPK, Rho GTPases, and PI3K/Akt signaling [19,20].

Fibrosis is described as the growth of fibrous connective tissue as a reparative response to injury or damage. Fibrosis could result from excess tissue deposition brought on by a pathological condition, or connective tissue accumulation that takes place during normal healing. Tissue healing processes such as fibrosis are associated with type II EMT. Type II EMT generates myofibroblasts from epithelia to heal damaged tissues; if the injury is moderate and acute, the healing event is referred to as reparative fibrosis. Therefore, tissue fibrosis is an unceasing sort of wound healing resulting from an abnormal inflammatory process [21].

Natural products have been recognized for centuries as an alternative source of natural therapeutic agents in the treatment of wound healing due to their effective application [22,23]. Recently, interest in olive oil and its by-products has increased [24]. Olive had been confirmed to have several benefits, such as the capability to modify the structure of neurotoxic proteins which are related to the debilitating effects of Alzheimer’s disease, along with the ability to delay and diminish the proliferation of cells, fibrogenesis, invasiveness, and EMT in several types of cancer including breast, kidney, and prostate cancers [13]. Hydroxytyrosol (HT) is a phenolic compound extracted from the olive tree as a by-product of olive oil production. It is regarded as the strongest antioxidant compound after gallic acid, and it is one of the most powerful antioxidant compounds among phenolic compounds from the olive tree, followed by oleuropein, caffeic acid, and tyrosol [25]. HT possesses significant free-radical scavenging activity and protects against oxidative stress. The antioxidant mechanism of HT is interrelated to its high degree of absorption and bioavailability after ingestion in the intestine [26]. In addition to its antioxidant activity, HT has many beneficial properties such as anti-inflammatory, anticancer, skin and eyes protector, diminished cell invasiveness, fibrogenesis and EMT in various types of cancers. Moreover, because of its capacity to lower blood pressure, enhance lipid profiles, and inhibit the formation of atherosclerotic lesions, HT will be an effective cardioprotective molecule [27].

Many studies explore the effect of HT towards different types of cells [28]. However, studies have yet to be undertaken about the HT effect on EMT type II in human dermal fibroblasts. Thus, this study intended to explore the effects of HT on TGF-β induced EMT type II in human dermal fibroblasts (HDF).

## 2. Results

### 2.1. HT Increases HDF Proliferation

To evaluate the HT effect towards cells viability and proliferation, HDF were treated with different concentrations of HT (0, 0.025, 0.05, 0.1, 0.15, 0.2, 0.4. 1 and 2%) for 24 h. Figure 1 shows the effect of HT on HDF viability after 24 h of treatment. The result showed that HT supplementation in all concentrations from all the samples (n = 5) promoted the proliferation of HDF. This result suggests that HT in all concentrations had no toxicity effect on HDF cell viability and proliferation after 24 h of incubation. Moreover, HDF proliferation was significantly increased as compared to the control group, with HT concentrations at 0.2% and 0.4% by (31.3 ± 6.9%) and (36 ± 15.4%), respectively, where *p*-value < 0.05, as shown in Figure 1. As a result, HT at concentrations of 0.2% and 0.4% were selected for further experiments.

### 2.2. Cell Morphology

Cultured HDF in FD medium, (FD + TGF-β), (FD + TGF-β + 0.2% HT) and (FD + TGF-β + 0.4% HT) showed a spindle shaped appearance and proliferated well in all groups’ media after four days of culture, as seen in Figure 2. Moreover, HDF cells cultured with 0.2% and 0.4% HT were found to be actively proliferated more than cells in the TGF-β group, which did not reach confluence after four days from seeding and tended to be flatter in shape than cells in control or HT supplementation groups. There was no difference in confluency between control group and cells treated with HT in both concentrations.

### 2.3. HT Enhances Cell Migration

In vitro scratch assay was performed to measure the rate of HDF cell migration in the presence of HT, and images of the scratched cells are shown in Figure 3. Based on the results, it was found that the HDF treated with 0.4% HT significantly increased the cell migration rate by (74.34 ± 34%) where *p* < 0.05 compared to TGF-β after 24 h of scratch induced as shown in Figure 4. There was no significant difference between cell migration rate cells treated with 0.2% HT and TGF-β group 24 h post injury. After 48 h, cells cultured with 0.2% and 0.4% HT showed a significantly higher migration rate (33.62 ± 11.94%, *p* < 0.05), as shown in Figure 4, and a complete wound closure earlier than cells in TGF-β group, whereas part of the wound area was still visible as displayed in Figure 3. There was no significant difference between cells cultured with 0.2% and 0.4% and cells cultured in FD (control) in cell migration rate.

### 2.4. Transwell Migration Assay

A transwell migration assay was carried out to determine whether the HT could exert a migration effect on HDF. Figure 5 shows the representative pictures that were taken during cell migration. The result showed that, after 48 h of incubation, HDF cultured with 0.2% and 0.4% HT significantly enhanced the migration capacity of HDF by (125.2 ± 14.52%) and (149.1 ± 19.24%), respectively, towards the media in the lower chamber as compared to the TGF-β group and control group (*p* < 0.05), as shown in Figure 6. This result suggests that HT can enhance cell migration capacity.

### 2.5. Immunocytochemistry Analysis

The immunoreactivities of E-cadherin and vimentin on the monolayer HDF cultured in FD, (FD + TGF-β), (FD + TGF-β + 0.2%HT), and (FD + TGF-β + 0.4%HT) were evaluated after 5 days. The immunopositivity of E-cadherin and vimentin can be seen in variable proportions. E-cadherin expression revealed moderate immunopositivity of approximately 41–70% in cells cultured with (FD + TGF-β + 0.2%HT) and (FD + TGF-β + 0.4%HT) compared to the control (FD) and (FD + TGF-β) groups, where some of the HDF cells did not express it, as shown in Figure 7. In contrast, the expression of vimentin demonstrates mild immunopositivity of almost 6-40% in cells cultured in (FD + TGF-β + 0.2%HT) and (FD + TGF-β + 0.4%HT) compared to the control (FD) and (FD + TGF-β) groups, as shown in Figure 7. The semi-quantitative analysis of staining for E-cadherin and vimentin indicates that HT up-regulates the expression of E-cadherin and down-regulates the expression of vimentin. The semi-quantitative scoring is summarized in Table 1.

## 3. Discussion

Unceasing inflammation in the wound healing process leads to pathological wound healing, resulting in fibrosis and scar formation. These pathological changes are mainly due to the EMT process, where cells lose their epithelial features and become mesenchymal cells [21]. This event can be investigated by evaluating the immunoreactivity of epithelial protein markers such as E-cadherin and mesenchymal protein markers such as vimentin. The epithelial adherence junction largely relies on E-cadherin. Reduced E-cadherin protein expression marked the start of EMT, in which cells began to separate from one another and migrate [29]. In the meantime, vimentin is a crucial mesenchymal marker that is significantly increased during EMT, indicating changes in the cell phenotype from epithelial to mesenchymal [30]. In this study, we investigated the HT effect to prevent TGF-β induced EMT in HDF.

Many natural compounds have been identified to be helpful in reducing inflammation and fibrosis by targeting specific inflammatory cytokines and signaling pathways. During the inflammatory phase, the nuclear factor kβ (NF-kβ) initiates the transcriptional process of several cytokine genes [31]. HT has anti-inflammatory action, as it targets the inflammatory cytokines such as TNF-α, IL-6, IL-8, and IL-1β. HT suppressed the TNF-α-induced activity of the nuclear factor of kappa light polypeptide gene enhancer in B-cells inhibitor, alpha (IκBα) and nuclear factor kappa-B kinase subunit beta (IKKβ). This leads to the stopping of the NF-κβ signaling pathway and terminates the production of the inflammatory cytokine which was marked by the down-regulation of the chemokine (CAC) motif ligand 2 (CCL2), and prostaglandin-endoperoxidase synthase 2 (PTGS2) [32].

It has been observed that all HT concentrations were nontoxic to HDF in the MTT assay. Moreover, all concentrations increased the cell viability above 100%. This is consistent with a previous study, which showed that HT in concentrations up to 30 µMwas nontoxic to HDF [33]. Another study, carried out by Cheng et al. (2017) on Human Umbilical Vein Endothelial cells (HUVECs), also demonstrated that HT at 30 µM promotes cell proliferation [34]. Treatment with HT did not alter or change the morphological appearance of HDF in any concentration. MTT assay results also revealed that HT at 0.2% and 0.4% concentrations were the best two concentrations that increased HDF viability, and were chosen to be used for the subsequent experimental assays for studying cell migration and proteins expression.

Regarding cell morphology, HT supplementation in culture was able to preserve the morphological characteristic of HDF. HDF cells showed the appearance of their typical spindle shape. Indeed, HDF were matured in monolayer culture, and cell proliferation increased as soon as passage one. This study confirmed that the supplementation of HT in culture was able to maintain the cell’s shape, which means that HDF maintains their phenotype with growth acceleration. A study carried out by Avola et al. (2019) supported the theory that HT protects HDF from damage caused by blue light and can maintain their morphology [35].

Regarding cell migration and motility, two assays were used in this study to evaluate the HT effect on HDF in wound healing with the induction of EMT: scratch assay and transwell migration assay. However, after 48 h, HDF treated with 0.2% and 0.4% completely closed the wound, whereas cells in the TGF-β group still did not fully close the wound. This indicated that adding HT to HDF culture media in the presence of an EMT induced model (5 ng/mL of TGF-β) enhances cell migration and proliferation, which are very important in the repairing of the wound process, which might be related to anti-inflammatory effects and angiogenesis modulation of HT. Previous studies in Swiss mice also showed that fibroblasts treated with olive oil revealed increased cell migration and improved wound repair [36,37]. In a transwell migration assay, it has been observed that HDF cultured with 0.2% and 0.4% HT had increased the migratory capacity of HDF more than the TGF-β group and the control group after 48 h of incubation. No previous study was carried out using a transwell cell migration assay regarding the effect of HT in HDF to evaluate the migration capacity either in human cells or animal models. However, many studies carried out on HUVECs stated that HT stimulates the migratory activities of HUVECs by promoting and activating the proto-oncogene tyrosine-protein kinase Src (Src) gene, which, along with the RhoA/Rho-associated protein kinase (ROCK) pathway, is crucial for the migration of endothelial cells and proper tubulogenesis [34,38].

It has been well documented that EMT E-cadherin will be down-regulated and vimentin will be up-regulated. Our study showed no difference in the immunoreactivity of E-cadherin in cells treated with HT of monolayer culture. E-cadherin was expressed at the same moderate immunostaining intensity in HDF cultured in 0.2% and 0.4% HT, and this could be due to the up-regulation of E-cadherin in these groups. However, the decrease in the protein expression level of vimentin in monolayer HDF cultured in 0.2% and 0.4% HT compared to control and TGF-β groups had backed that the supplementation of HT was able to suppress the EMT process, indicating by down-regulation of the mesenchymal marker vimentin, and could enhance the wound repair without any pathological healing. There was no difference in the immunoreactivities of the vimentin in the control and TGF-β group. A previous study carried out by Razali et al. (2018) on the effect of HT in protein expression on RECs supports our result that HT inhibits EMT by preventing the activation of TGF-β signaling, with up-regulation E-cadherin and down-regulation vimentin [39].

## 4. Materials and Methods

### 4.1. Human Dermal Fibroblasts Isolation and Culture

Isolation and ethical approval were obtained from the Research and Ethical Committee of the Faculty of Medicine, Universiti Kebangsaan Malaysia (FF-2022-014). All the human study subjects provided informed consent. Human dermal fibroblasts (HDF) were obtained from five consenting patients who underwent abdominoplasty or face-lift surgery, with no specificity regarding gender or age groups. Skin samples (3 cm^2^) were cleaned of undesirable fragments, for example fat, hair, and debris, and then were minced into small pieces (about 2 mm). The skin was digested with 0.6% Collagenase Type I (Worthing- ton, Lakewood, NJ, USA) for 5–6 h in a 37 °C incubator shaker, followed by the dissociation of the cells using 0.05% Trypsin-EDTA (Elabscience, Houston, TX, United State of America (USA)) for 8–10 min. Digested skin containing both keratinocytes and fibroblasts was then re-suspended in culture medium (fibroblasts growth medium, i.e., Dulbecco’s Modified Eagles Medium F12 (DMEM/F-12, Elabscience, Houston, TX, USA) supplemented with 10% FBS (Sigma-Aldrish, Missouri, MO, USA), and were seeded into three wells (surface area of 9.6 cm^2^/well) of a six-well culture plate (Greiner Bio-One, Monroe, NC, USA) at 37 °C in 5% CO_2_. Waste media were changed every two to three days. When cells reached 70–80% confluence, fibroblasts were detached from the culture surface using 0.05% trypsin–EDTA for a maximum of 5 min to separate the fibroblasts from the culture surface. Detached fibroblasts were then re-cultured in a T75 flask (Greiner Bio-One, NC, USA) with DMEM/F12 containing 10% FBS [40]. The media were changed every two days until the cells reached 80–90% confluency before being trypsinized into passage 1 or passage 2, which were used as the experimental passages.

The morphological features of HDF were assessed daily under an inverted light microscope (Nikon, Tokyo, Japan). Cell viability and the proliferation rate for each sample were evaluated following five days of culture.

### 4.2. Experimental Design

The study design is an in vitro study that aims to evaluate the effect of HT on the viability, morphology, migration, and protein expression of HDF in the EMT model. The best two concentrations of HT were chosen based on the maximum cell viability shown by the effect of HT in the MTT assay. For migration assay and proteins expression, the HDF culture was divided into four groups: control which HDF cultured in media only (FD), EMT induced (FD+TGF-β), treated with 0.2% HT (FD + TGF-β + 0.2% HT), and treated with 0.4% HT (FD + TGF-β + 0.4% HT).

### 4.3. Hydroxytyrosol

HT (3,4-dihydroxyphenylethanol, HT) was purchased from Sigma Aldrish (Sigma, St. Louis, MO, USA). The bottle has 25 Mg in liquid form. The purity of HT is ≥98% with the formula C8H10O3.

### 4.4. MTT (3-(4,5-Dimethylthiazol-2-yl)-2,5-diphenyltetrazolium bromide) Assay

The cytotoxic effects of commercially available HT (Sigma, St. Louis, MO, USA) were evaluated using (3-(4,5-Dimethylthiazol-2-yl)-2,5-Diphenyltetrazolium Bromide) MTT assay (Invitrogen^TM^, Carlsbad, CA, USA) following the manufacturer’s instructions. Briefly, human dermal fibroblasts in passage one were cultured in triplicate using a 96-well plate at a density of 10,000 cells/cm^2^ in culture media that contained 10% FBS and 1% penicillin/streptomycin for 24 h. After 24 h of incubation, the media were replaced and the cells were subjected to several concentrations of HT (0, 0.025, 0.05, 0.1, 0.15, 0.2, 0.4. 1 and 2% (*v*/*v*)) [39]. After 24 h post-treatment, the MTT assay was carried out. During the assay, HT-treated fibroblast media were discarded and replaced by 90 µL of DMEM with 10 µL of MTT solution and incubated for four hours at 37 °C. Then, 100 µL of dimethyl sulfoxide (DMSO, Sigma Aldrish) solution was added to release and solubilize the purple formazan and it was further incubated for 15 min. After that, the absorbance was read at 540 nm. Five samples were assessed in triplicate for this assay. The two best concentrations were chosen to be used for further experimentation.

### 4.5. EMT Model Induced by Adding TGF-β

With the purpose of inducing an EMT model, 5ng/mL TGF-β was added to the human dermal fibroblasts in treated and non-treated cells with HT based on previous studies [39,41,42].

### 4.6. Scratch Assay

Scratch assay was performed to evaluate the wound healing effect of HT on HDF. Cells were seeded into 12-well plates at a density of 12 × 10^4^ cells/well and incubated for 48 h until they reached 90% confluency in 37 °C and 5% CO_2_ incubator [43]. Media were removed, and the cell monolayers were washed two times with 1 mL of PBS and one time with DMEM. Then, the monolayer cells were scratched using a 200 µL sterile pipette tip to make a wound and washed two times with PBS to wash away the cell’s debris, then incubated with serum free media with HT at different concentrations. The wound closures were observed, and pictures were taken at 0, 24, and 48 h. Results were analyzed using Image-J software and closed area percentages were measured and compared with the EMT-induced group (FD + TGF-β). An increase in closed area percentage indicates the migration of HDF cells.

### 4.7. Transwell Migration Assay

Transwell migration assay was conducted using Falcon^®^ Permeable Support with 4.0 µm PET membrane (Corning, New York, NY, USA) in the presence of culture media (containing 10% FBS) as a chemoattractant, which was added to each well of the 6-well plate. The cell culture inserts were aseptically placed into each well. After the cells were trypsinized and counted, the cells were resuspended in serum free media with different treatments at a seeding density of 1 × 10^5^ cells/0.4 mL for each of the inserts, and the migration of the cells toward the bottom of the insert was allowed to occur during the incubation of the well plate for 48 h at 37 °C and 5% CO_2_. After 48 h of incubation, non-migrated cells in the upper chamber were softly removed using small cotton wipes. Then, the migrated cells at the bottom of the insert were fixed with 1mL 4% paraformaldehyde (Elabscience, TX, USA) for each well and washed thrice with PBS. Then, the cell culture inserts were incubated in the dark with 1ml 4′,6-diamidino-2-phenylindole (DAPI; Thermo Fisher Scientific, Carlsbad, CA, USA) at 1:15,000 dilution for each well for 40 min at room temperature, and then were washed thrice with PBS. Pictures of DAPI-stained nuclei at the bottom of the inserts were taken at five different fields using an immunofluorescence microscope (Nikon, Tokyo, Japan) before the numbers of nuclei were averaged [44].

### 4.8. Immunocytochemistry Analysis

Immunocytochemical analysis was carried out to evaluate the expression level of E-cadherin and vimentin. First, cells were fixed with 4% paraformaldehyde (PFA) (Elabscience, USA), 200 µL each well, for 15 min. Then, cells were washed two times with 500 µL of PBS each time for 5 min, permeabilized for 20 min with 200 µL of 0.5% Triton X-100 solution (Sigma-Aldrich), and then blocked with 500 µL of 10% goat serum (Elabscience, USA) for 1 h at 37 °C. The cells were then incubated with 1:200 mouse anti-E-cadherin antibody and 1:200 rabbit anti-Vimentin antibody (Elabscience, USA) overnight at 4 °C. On the following day, the mixture solution was discarded, and cells were washed three times with 500 µL PBS before being incubated with 1:500 diluted Alexa Fluor 594 anti-rabbit IgG (Abcam, Cambridge, MA, USA) and 1:300 Alexa Fluor 488 anti-mouse (Abcam, Cambridge, MA, USA) for 2 h at 37 °C. Nuclei were counterstained with 200 µL of DAPI for 20 min at room temperature in the dark. The plate was viewed under an immunofluorescent microscope (Nikon A1, Nikon Corporation, Tokyo, Japan) in the dark, and fluorescent images were captured at 10 different fields. Simple qualitative scoring was applied to the monolayer HDF culture taken from Munirah et al. (2010) [45].

### 4.9. Statistical Analysis

The data were analyzed using GraphPad Prism version 8.0 for windows (GraphPad Software, Inc., San Diego, CA, USA). Averaged data were represented as mean ± standard error of mean (SEM). One-way analysis of variance (ANOVA) was used to analyze the data for MTT assay, scratch assay and transwell migration assay. Statistical significance was set at a value of *p* < 0.05.

## 5. Conclusions

Our findings suggests that HT can inhibit EMT, potentially inhibiting fibrosis and persistent inflammation related to skin wound healing. Further studies using in vitro disease models are needed to support our conclusion and findings about the effect of HT on EMT in tissue repair and wound healing.

## Figures and Tables

**Figure 1 molecules-28-02652-f001:**
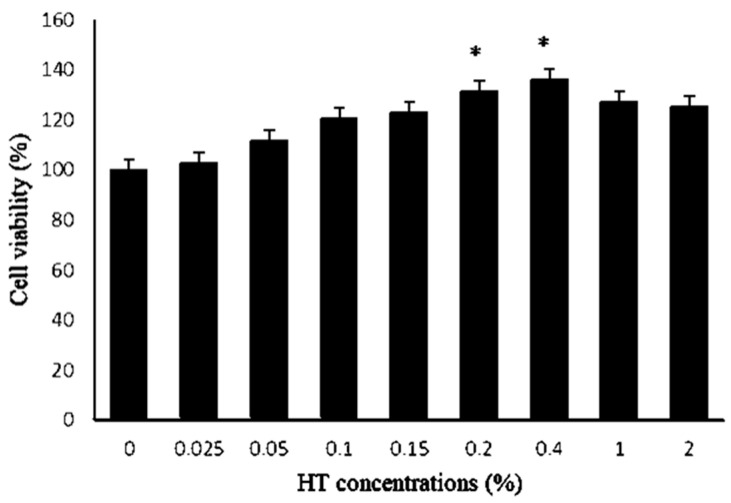
The MTT assay analysis. HDF was cultured in several concentrations of HT in the culture medium. HDF cultured in HT at concentrations of 0.2 and 0.4% showed an increased amount of cell viability by 31.3 ± 6.9% and 36 ± 15.4%, respectively, as compared to control where * *p* < 0.05. The values were expressed as mean ± SEM, n = 5.

**Figure 2 molecules-28-02652-f002:**
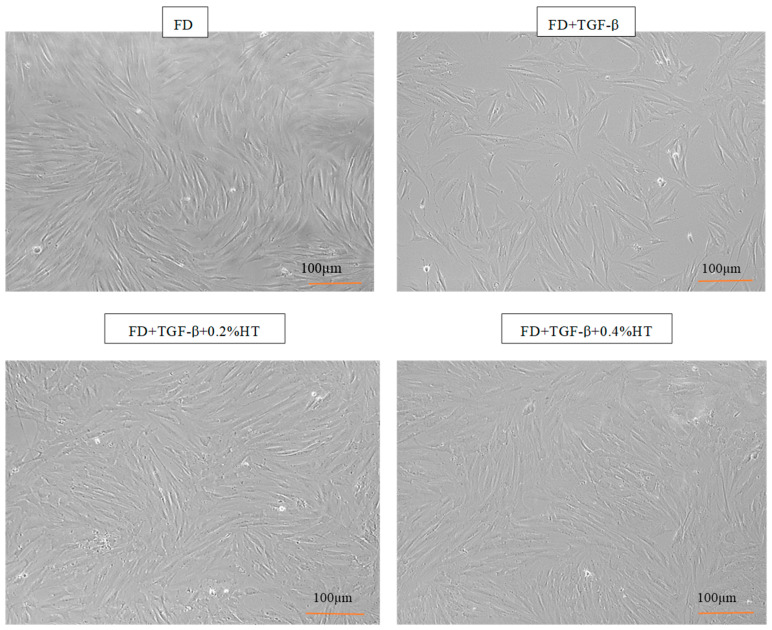
Morphology of HDF after 4 days of culture in all media groups at 10× magnification. After 4 days, HDF maintain their spindle shapes in all experimental groups. HDF cells cultured with 0.2% and 0.4% HT were found to be actively proliferated more than cells in the TGF-β group. Scale bar = 100 µm.

**Figure 3 molecules-28-02652-f003:**
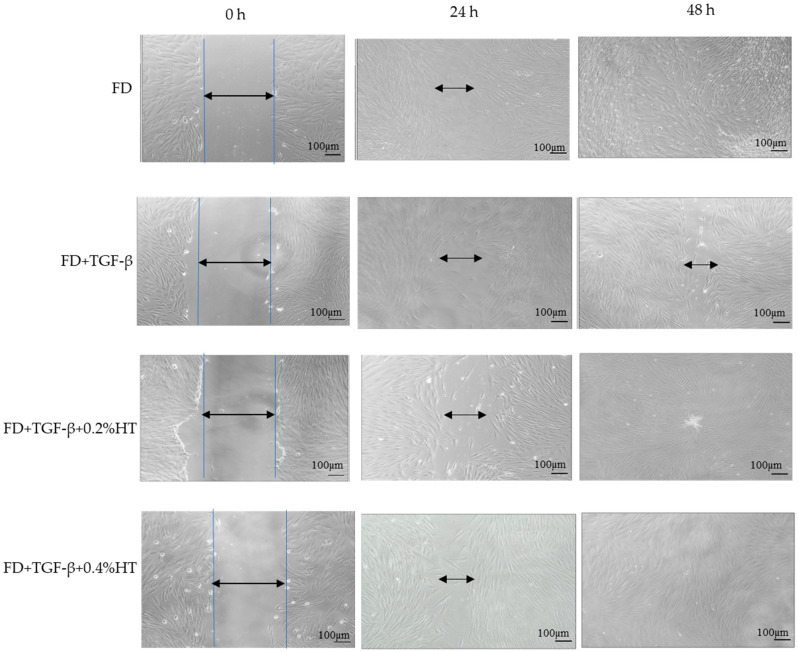
Representative pictures show the migration of HDF cells at magnification ×10 after induction of a scratch. Pictures illustrate that cells treated with HT show a higher migration rate compared to the TGF-β group. All the pictures were taken immediately after the scratch was induced (0 h), and after 24 h and 48 h. Scale bar = 100 µm.

**Figure 4 molecules-28-02652-f004:**
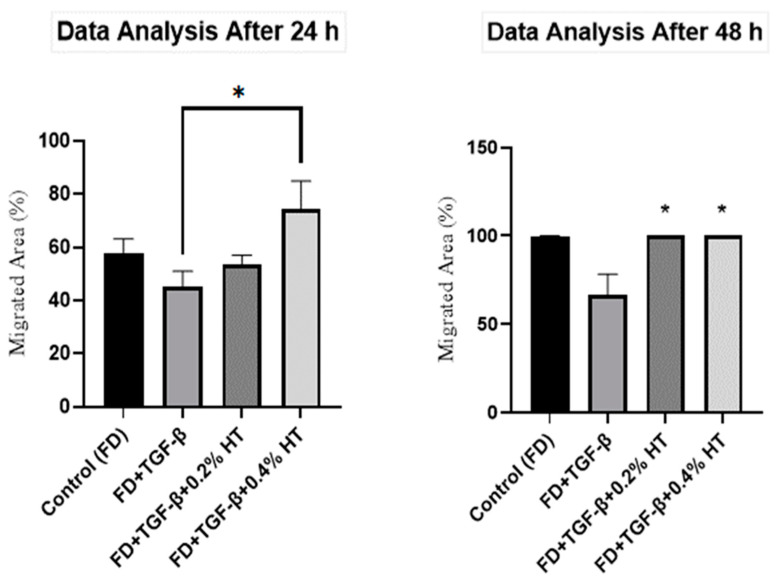
Scratch assay analysis for 24 h and 48 h. The graph shows that the cells’ migration rate in FD+TGF-β+0.4% HT was significant (* *p* < 0.05) as compared to other groups. After 48 h, cells cultured in both 0.2% and 0.4% significantly increased migration rate as compared to FD+TGF-β (* *p* < 0.05). Values represented as mean ± SEM, n = 5.

**Figure 5 molecules-28-02652-f005:**
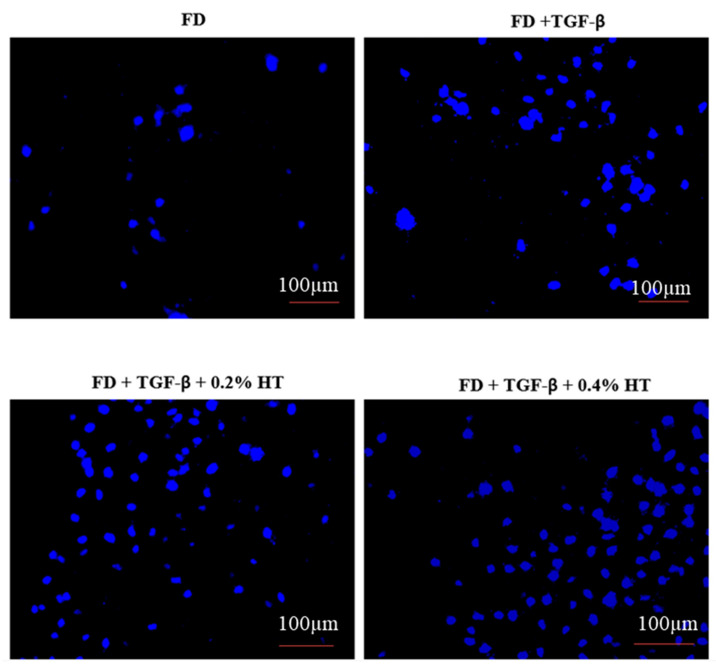
The transwell migration assay. Representative images of migrated HDF at 10× magnification, and the nuclei stained with DAPI at the bottom of transwell inserts. Scale bar = 100µm.

**Figure 6 molecules-28-02652-f006:**
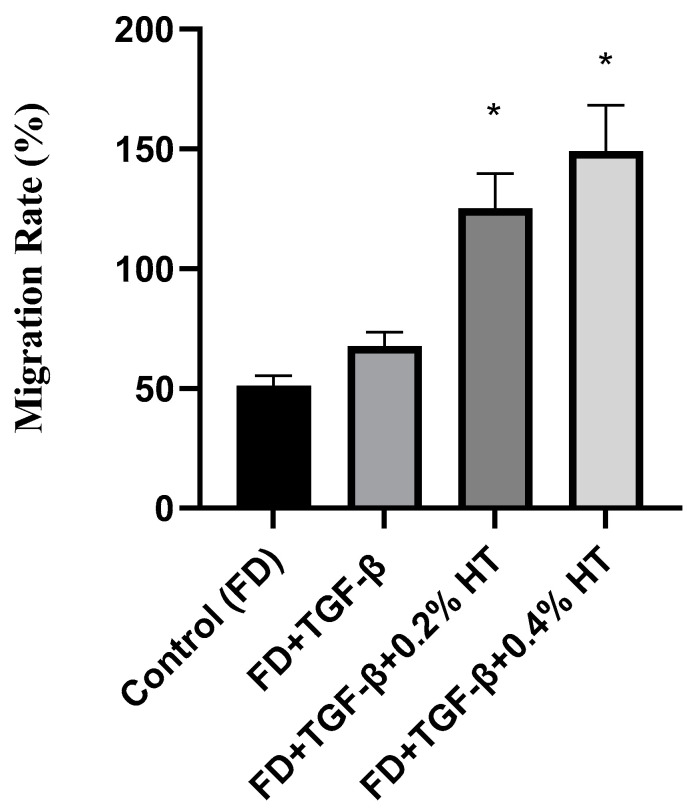
Transwell migration assay. The graph shows that HDF cultured in 0.2% and 0.4% HT significantly enhance the migration rate in comparison to control and FD + TGF-β (* *p* < 0.05). Value represented as mean ± SEM, n = 5.

**Figure 7 molecules-28-02652-f007:**
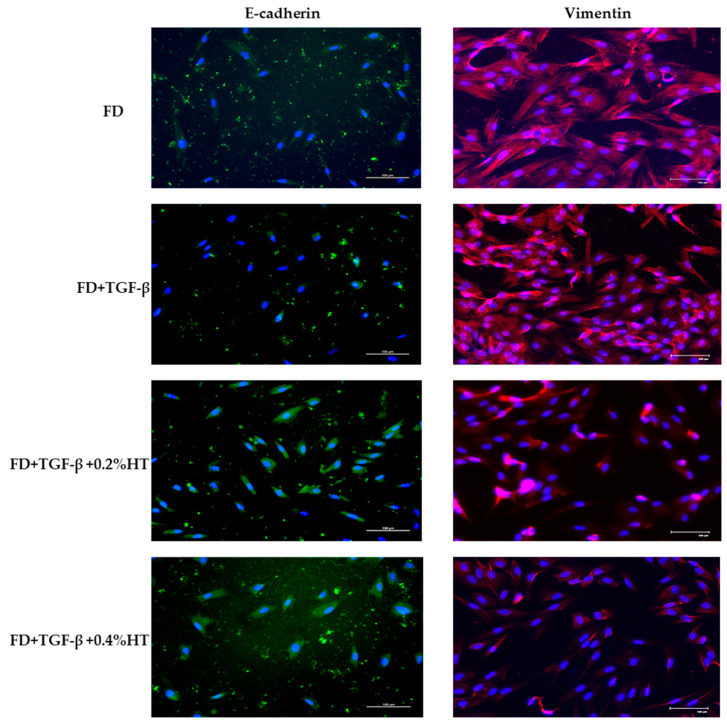
Immunocytochemistry staining of all experimental groups. A moderate staining for E-cadherin was observed in cells treated with 0.2% and 0.4% HT after 5 days of culture. Vimentin staining is weaker in cells treated with 0.2% and 0.4% HT; however, it is moderately stained in various proportions (41–70%) in both control and TGF-β groups. Green = FITC, red = Texas Red, blue = DAPI (nucleus staining). Scale bar = 100 µm.

**Table 1 molecules-28-02652-t001:** Semi-quantitative scoring of immunocytochemistry of HDF treated with/without HT. The colors were assessed using the semi-quantitative score (- up to +++) representing the estimated percentage of the stained cells observed under immunofluorescent microscope (+: 6–40%, ++: 41–70%,).

Culture Medium	E-Cadherin	Vimentin
Control (FD)	+	++
FD + TGF-β	+	++
FD + TGF-β + 0.2%HT	++	+
FD + TGF-β + 0.4%HT	++	+

## Data Availability

Not applicable.

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
