# Peer review of "The Effect of Hydroxytyrosol in Type II Epithelial-Mesenchymal Transition in Human Skin Wound Healing"

_molecules, 2023, doi:10.3390/molecules28062652_

Round 1

Reviewer 1 Report

This study showed that Hydroxytyrosol (HT) has ability to inhibit EMT which has potential in inhibition of fibrosis and persistent inflammation related to skin wound healing. This study seem interesting, but there are some comments to be raised.

- Figure 1, the concentration range (0.025-2%), It's more appropriate to convert concentration to either ug/ml or uM

- Figure 2, microscopic figures should be more informative by adding arrows and signs 

- Please, add scale bar for all the microscopic images

- The error bar in Bar representation of figure 5 are high values that the mean. results should be revised.

- It will be well suggested adding molecular docking of Hydroxytyrosol (HT) towards the tested target

- Plagiarized parts should be parphrased especially in the experimental section

- Statistical analyses should be well-revised

Author Response

We appreciate the time and efforts by the editor and the reviewers in reviewing this manuscript. We addressed all the issues indicated in the review report. This manuscript has been amended following Reviewer 1 comments and suggestions. Hopefully, the revised version of the manuscript was able to address the reviewer’s concern and meet the journal publication requirements.

Reviewer 2 Report

I have read the text “The Effect Of Hydroxytyrosol In Type II Epithelial-Mesenchymal Transition In Human Skin Wound Healing”, and however the topic is interesting, the manuscript contains some drawbacks that must be addressed before possible final decision of the Editor to accept or not.

Drawbacks:

-          Line 22: Scratch assay showed increased cell migration in cells treated with HT (p<0.05) compared to TGF-β group. That sentence is not entirely right. There is difference between two HT doses when time “after 24 h” is considered. Similar effect of two dosages was after 48 h.

-          Figure 2 and description of photos: yes, I agree that all treatments maintain their spindle shapes but there is quite visual difference between control and FD+TGF-β. Some another description?

-          Line 295: add city name.

-          For example line 356: add city and US state. Check another and improve.

-          M&M: the source, purity and characterization of HT is missing. Add the appropriate info.

-          Discussion and conclusions: some possible mechanisms towards reduced inflammation and fibrosis state must be added.

-          The reference section must be checked and rewritten. The lack of paramount information (e.g. names of journals).

Author Response

We appreciate the time and efforts by the editor and the reviewers in reviewing this manuscript. We addressed all the issues indicated in the review report. This manuscript has been amended following Reviewer 2 comments and suggestions. Hopefully, the revised version of the manuscript was able to address the reviewer’s concern and meet the journal publication requirements.

Round 2

Reviewer 1 Report

The authors have addressed all raised comments. No further comments

Author Response

Dear Reviewer 1, Thank you.

Reviewer 2 Report

Although the authors stated that the reference section has been rewritten I cannot see those changes. Names of journals are missing in many places.

The rest of queries has been addreessed.

Author Response

(The authors gave the same response as above.)
